# Impact of Surgical Approach, Patient Risk Factors, and Diverting Ileostomy on Anastomotic Leakage and Outcomes After Rectal Cancer Resection: A 5-Year Single-Center Study

**DOI:** 10.3390/medicina61101751

**Published:** 2025-09-25

**Authors:** Deividas Nekrosius, Edvinas Gvozdas, Gabriele Marija Pratkute, Algimantas Tamelis, Paulius Lizdenis

**Affiliations:** 1Department of Surgery, Hospital of Lithuanian University of Health Sciences, LT-50161 Kaunas, Lithuania; algimantas.tamelis@lsmu.lt (A.T.); paulius.lizdenis@lsmu.lt (P.L.); 2Faculty of Medicine, Lithuanian University of Health Sciences, LT-44307 Kaunas, Lithuania; edvinas.gvozdas@stud.lsmu.lt (E.G.); gabriele.marija.pratkute@stud.lsmu.lt (G.M.P.)

**Keywords:** rectal cancer, anastomotic leakage, laparoscopic surgery, postoperative complications, risk factors, ASA score, NOAC, surgical outcomes

## Abstract

*Background and Objectives*: This study aimed to evaluate surgical outcomes and identify prognostic factors associated with anastomotic leakage (AL), following rectal cancer resection. *Materials and Methods*: A retrospective cohort study included 415 patients who underwent rectal cancer surgery between 2020 and 2024. Patients were categorized by surgical approach (laparoscopic vs. open) and presence of AL. *Results*: Of the 415 patients, 160 (38.6%) underwent laparoscopic surgery, and 255 (61.4%) underwent open surgery. Operative time was significantly longer for laparoscopic surgery (213.0 ± 65.9 vs. 201.3 ± 60.4 min, *p* = 0.05), while stoma formation was more frequent in the open surgery group (60.0% vs. 48.1%, *p* = 0.018). Reoperation rate was higher in the laparoscopic group compared to the open group (13.1% vs. 6.7%, *p* = 0.027). The rate of AL was 20.5% in the laparoscopic group and 18.4% in the open surgery group (*p* = 0.434). Patients with AL had a significantly longer hospital stay (17 days, IQR 12.0–23.7 vs. 8 days, IQR 7.0–9.0, *p* < 0.001). The use of NOACs was associated with an increased risk of AL (*p* = 0.026). Multivariate analysis revealed that both a higher ASA score (*p* = 0.022) and older age (*p* = 0.044) were independent risk factors for AL, while the use of a diverting ileostomy was associated with a threefold reduction in the risk of AL (*p* = 0.049). *Conclusions*: AL rates were similar between approaches. Laparoscopic surgery had more reoperations and longer operative times. AL was associated with NOAC use, older age, and higher ASA scores. Diverting ileostomy reduced AL risk and warrants broader use in high-risk patients to improve outcomes.

## 1. Introduction

Colorectal cancer is the second most common cancer in Europe with over 500,000 new diagnoses made annually [1]. Statistically, about one-third of colorectal cancer cases are located in the rectum with a similar number of cases being diagnosed at advanced stages [2].

The surgical approach is fundamental for cancer treatment [3]. Surgery is useful in both palliative and curative care; therefore, there is a great need to improve the understanding of possible outcomes and prognostic factors [4]. The continuous progress in this field introduces greater complexity in treatment selection [5]. The preferable choices for early rectal cancer are either removal by local excision or chemoradiotherapy followed by local excision. If a complete response is confirmed using only chemoradiotherapy, then a “watch and wait” strategy is optional [6]. However, advanced rectal cancer typically requires a radical surgical approach, determined by the tumor’s size and exact location, along with combined neoadjuvant chemoradiotherapy. This approach can increase the likelihood of preserving the anal sphincter by reducing the size and stage of the tumor, thereby securing a more adequate circumferential margin [7].

One of the biggest concerns of patients dealing with low rectal cancer is permanent stoma, which influences the surgical decision to extend into intersphincteric space and perform anastomosis [8]. Laparoscopic surgery can also be useful for operable low rectal tumors yielding minimal postoperative complications, low local recurrence rates and reduced incidence of low anterior resection syndrome [9]. Moreover, Laparoscopic surgery was associated with shorter hospital stays and reduced opioid use. Compared to laparoscopic procedures, open surgery significantly increased the likelihood of blood transfusion, ICU admission, and mechanical ventilation [10].

There is a spectrum of factors that collectively determine the success of rectal cancer surgery (intervention and following patient quality of life) and can alter postoperative outcomes [11]. Common short-term postoperative complications following rectal cancer surgery include anastomotic leaks, infections, bleeding, and wound dehiscence [12]. These complications can lead to heightened morbidity, extended hospital admissions, and the requirement for further interventions [13]. Postoperative complications could worsen the prognosis as well. Cytokines released by inflammation caused by either anastomotic leakage, abdominal abscess or pneumonia can cause tumor progression or metastasis. In addition, recovery from postoperative complications takes time, often delaying the initiation of adjuvant chemotherapy [14].

Surgical outcomes in rectal cancer are intrinsically tied to a range of patient-specific and tumor-related factors. The most recognized risk factors include the age and male gender of the patient, poor blood supply to the anastomosis, excessive tension, preoperative chemoradiotherapy, obesity, emergency surgery and coexisting medical conditions [15]. Several modifiable prognostic factors include timely neoadjuvant treatment, high surgeon volume, high tie of the inferior mesenteric artery, extension of mesorectal excision, and improved techniques for intersphincteric resection and anastomosis [16]. The use of protective ileostomy remains debated, as its leak-reducing benefits must be weighed against significant stoma-related morbidity, including dehydration, wound complications, and reoperation after reversal [17].

This study aimed to compare laparoscopic and open surgical techniques and identify prognostic factors associated with anastomotic leakage following rectal cancer resection.

## 2. Materials and Methods

### 2.1. Study Design

A retrospective data analysis was conducted in Hospital of Lithuanian University of Health Sciences Kaunas Clinics. The study period covered all rectal cancer resections performed between January 2020 and December 2024. The study was ethically sanctioned by the Bioethics Centre under the reference number 2024-BEC2-272.

### 2.2. Selection of Patients

This study considered for inclusion all patients who underwent rectal cancer surgery. Both elective and urgent surgical cases were eligible for inclusion. Exclusion criteria included patients undergoing palliative procedures without resection, endoscopic or transanal procedures; those with recurrent disease; and patients with incomplete clinical or pathological data. Patients were stratified according to the surgical approach (laparoscopic vs. open surgery) and postoperative outcomes (presence or absence of AL), Figure 1. AL was defined as an evident dehiscence of the anastomosis, diagnosed either during exploratory surgery or via abdominal computed tomography scan, and/or as the presence of a pelvic abscess occurring after rectal surgery and detected during the initial hospital stay.

### 2.3. Data Collection

Clinical, demographic and operative data were retrospectively collected from the hospital electronic medical records and pathology reports. The variables analyzed included age, sex, BMI, ASA classification, neoadjuvant treatment, tumor location, operative approach, intraoperative and postoperative details (e.g., type of operation, use of pelvic drain, formation of stoma, operative time) and pathological findings (e.g., ypT/ypN stage, lymph node yield). Postoperative outcomes such as length of hospital stay, reoperations, and anastomotic leakage were recorded. Patients with AL within 30 days postoperatively were identified through systematic monitoring of 30-day rehospitalizations. All patients operated on in our hospital are routinely referred to our tertiary center in case of early postoperative complication; the severity of these complications was evaluated by Clavien–Dindo classification, Table 1. Additional variables included comorbidities (arterial hypertension, diabetes mellitus) and use of anticoagulants (NOAC, aspirin, clopidogrel).

### 2.4. Data Analysis

Statistical analysis was performed with IBM SPSS Statistics for Windows, Version 29.0. Armonk, NY, USA. Continuous variables were tested for normality using the Kolmogorov–Smirnov test. Normally distributed data were expressed as mean ± standard deviation (SD) and compared using the independent samples *t*-test. Non-normally distributed data were reported as median and interquartile range (IQR) and compared using the Mann–Whitney U test. Categorical variables were expressed as frequencies (percentages) and compared using the chi-square test or Fisher’s exact test when appropriate. Results were considered statistically significant at *p* < 0.05. In the multivariable logistic regression model, we included five clinically relevant variables, with 36 anastomotic leak (AL) events available, corresponding to an events-per-variable ratio of approximately 7. To minimize potential overfitting, only clinically justified predictors were included and collinearity between variables was avoided.

## 3. Results

### 3.1. Patient Characteristics

A total of 415 patients who underwent rectal cancer resection surgery during the 5-year study period were included in this study. Among them, 160 patients (38.6%) underwent laparoscopic surgery, while 255 patients (61.4%) had open surgery. Clinical and demographic characteristics are detailed in Table 2. No significant differences were observed between groups in terms of age, sex distribution, BMI, ASA classification, neoadjuvant therapy, tumor distance from the anal verge, or anatomical tumor location. However, arterial hypertension was more prevalent in the laparoscopic group (68.8% vs. 58.1%, *p* = 0.03).

### 3.2. Intraoperative Outcomes

Operative time was slightly longer in the laparoscopic group compared to the open group (213.0 ± 65.9 vs. 201.3 ± 60.4 min, *p* = 0.05). Pelvic drains were more frequently placed in the open surgery group (97.2%) than in laparoscopic procedures (93.1%, *p* = 0.05). Additionally, stoma formation was significantly more common among patients undergoing open surgery (60.0% vs. 48.1%, *p* = 0.018) (Table 3).

### 3.3. Postoperative Outcome

Laparoscopic resection was significantly more frequently performed in patients with T1 stage tumors (11.9% vs. 2.9%, *p* = 0.007). Lymph node yield was significantly higher in open surgery cases (13.1 ± 7.8 vs. 11.2 ± 5.8, *p* = 0.01). Postoperative recovery was largely comparable between groups, with similar median hospital stays (9 days for both groups, *p* = 0.682). The incidence of anastomotic leak (AL) was not significantly different between laparoscopic (20.5%) and open surgery (18.4%) groups (*p* = 0.434). However, patients undergoing laparoscopic surgery required reoperation more frequently (13.1% vs. 6.7%, *p* = 0.027) (Table 4). Postoperative complications according to Clavien–Dindo classification are shown in Table 5. The overall complication rate was 41.8% in the laparoscopic group and 40% in the open group. Although the distribution of grades differed slightly between groups, there were no statistically significant differences in any complication category between laparoscopic and open surgery. (*p* = 0.237). However, the 30-day readmission rate was significantly higher after laparoscopy compared with open surgery (5.0% vs. 1.8%, *p* = 0.006).

### 3.4. Risk Factors for Anastomotic Leak

AL occurred in 36 cases (19.4%), Figure 2. Additional analysis of postoperative outcomes and prognostic factors associated with anastomotic leakage is presented in Table 6. Patients who developed AL had a significantly longer hospital stay compared to those without AL (median 17 days, IQR 12.0–23.7 vs. 8 days, IQR 7.0–9.0; *p* < 0.001). ASA score distribution differed significantly between the groups, with a higher AL rate of ASA IV patients (57.1%, *p* = 0.002). Additionally, patients with arterial hypertension tended to have a lower AL rate (14.4% vs. 25.6%, *p* = 0.05). The use of anticoagulants, particularly NOACs, was associated with an increased risk of AL (*p* = 0.026). The rate of diverting ileostomy was 20.9%. Multivariate analysis revealed that both a higher ASA score (*p* = 0.022) and older age (*p* = 0.044) were independent risk factors for anastomotic leakage (AL) (Table 7). Conversely, the use of a diverting ileostomy was associated with a threefold reduction in the risk of AL (*p* = 0.049).

## 4. Discussion

This retrospective study analyzed postoperative outcomes in patients undergoing laparoscopic and open rectal cancer surgery, with a focus on anastomotic leakage, reoperation rates, lymph node harvest and comorbidities.

The optimal surgical approach to minimize anastomotic leakage remains debated. A meta-analysis done by Zheng et al. [18] found that a laparoscopic approach reduces the incidence of anastomotic leakage because of better pelvis exposure. Another meta-analysis done by Arezzo et al. [19] found no significant differences in open versus laparoscopic technique used regarding anastomotic leakage (*p* = 0.128). The study by J. Lujan et al. [20] showed no significant differences in complications or reoperations between the open and laparoscopic surgery groups. In our study the overall anastomotic leakage rate did not differ significantly between surgical techniques (*p* = 0.78), while the reoperation rate was significantly higher in the laparoscopic group (*p* = 0.027). The greater reoperation rate in our cohort may reflect a more frequent use of primary anastomosis in laparoscopic cases.

We found that laparoscopic surgeries were significantly longer than open surgeries (213.0 ± 65.9 vs. 201.3 ± 60.35 min, *p* = 0.05), which is consistent with previous studies. A meta-analysis by Trastulli et al. [21] also reported longer operative times for laparoscopic surgery, although with high heterogeneity. The authors attributed this variability to differences in surgeons’ experience with laparoscopic techniques.

The number of harvested lymph nodes was significantly higher in open surgery (13.1 vs. 11.2, *p* = 0.01). This difference may be explained by the more extensive manual exposure and tactile feedback during open dissection. However, other studies have reached opposite conclusions. Boutros et al. [22] have proposed that laparoscopic surgery for rectal cancer may provide better visualization and more accurate pelvic dissection with less tissue manipulation. Improved retraction and easier access to the most proximal portion of the inferior mesenteric vessels contribute to greater harvested lymph nodes in the laparoscopic group, according to this study. On the other hand, Yamamoto et al. [23] conducted a similar study, but with patients who had a BMI over 25. They found that the median number of lymph nodes harvested in the laparoscopic group was significantly lower compared to the open surgery group (17.5 vs. 21.0 *p* = 0.0047). These results correlate with our study, where the median BMI of our cohort was over 25 in both the laparoscopic and open groups. These findings suggest that for patients with a BMI higher than 25, more lymph nodes can be harvested using an open surgery approach.

Fewer pelvic drains were placed in the laparoscopic group (91.9% vs. 94.4% *p* = 0.046). Pelvic drains can also increase postoperative pain and bowel obstruction due to their foreign body reaction, or act as a gateway for wound infection. Therefore, the routine use of pelvic drains remains controversial [24]. A study by Denost et al. [25] concluded that pelvic drainage did not decrease reoperation rates or detect early anastomotic leakage. However, surgeons should consider using pelvic drains in patients undergoing longer, higher-volume surgeries [26]. This is consistent with our findings, as significantly more patients with T1 tumors—typically requiring less extensive and shorter surgeries—were operated on laparoscopically. This could be the reason why pelvic drains were used significantly less frequently in the laparoscopic group.

Our study found several factors associated with anastomotic leakage. The ASA score is a useful method for patient performance status evaluation using multiple characteristics. Therefore, association between ASA scores and postoperative complication rates was observed [27]. According to a study by Bakker et al. [28] an ASA score of III–IV increased the odds of anastomotic leakage significantly. These findings correlate with our study, where we found that a significant number of patients with an ASA IV score experienced anastomotic leakage (57.1%). A study by Moon et al. [29] found that NOAC use was associated with significantly higher levels of anastomotic leakage, small bowel obstruction, infection, and renal complications. These findings correlate with our results.

Conversely, patients with arterial hypertension had a significantly lower leakage rate. Previous studies have often reported hypertension as a risk factor for surgical complications. Post et al. [30] found that high preoperative diastolic blood pressure and profound intraoperative hypotension, in addition to complex surgery marked by a blood loss of more than 250 mL, are associated with an increased risk of developing anastomotic leakage. However, it is possible that hypertensive patients in our cohort were more closely monitored and medically optimized pre- and postoperatively, resulting in more stable hemodynamic conditions and improved tissue perfusion contributing to improved short-term outcomes.

The anastomotic leakage rate observed in our cohort (19.4%) is relatively high compared to rates reported in large multicenter studies. In the COLOR II trial, leakage occurred in 13% of patients after laparoscopic and 10% after open surgery [31]. Similarly, the Dutch TME trial reported a leakage rate of 10.4% following low anterior resection [32] while the ALARM international study found an overall leakage incidence of 8.6% across multiple centers and countries [33]. The elevated rate in our study may be explained by a higher proportion of ASA IV patients, more frequent use of NOAC, and fewer diverting stomas. In our cohort, only 20% of patients received a diverting stoma, compared to approximately 35% in the COLOR II trial [31]. Additionally, our findings align with recent evidence from a large meta-analysis [34] and multi-institutional study [35], both of which demonstrate that diverting ileostomy significantly lowers the risk and severity of anastomotic leakage. These data, combined with our own, support a more proactive approach to fecal diversion in high-risk rectal cancer patients.

This study has several limitations. First, due to its retrospective design, it relied on the accuracy and completeness of existing medical records, which may have led to unrecorded confounding factors such as nutritional status, smoking history, or exact intraoperative decision-making processes. Secondly, while statistical associations were observed, the subgroup sample sizes, such as patients using NOACs or having arterial hypertension, were relatively small, potentially affecting statistical power. Third, the analysis was limited to short-term postoperative outcomes. Important long-term outcomes such as local recurrence, disease-free survival, and quality of life were not evaluated. Fourth, TME quality assessment was only implemented as a routine practice starting in 2022; therefore, data for preceding years are incomplete or missing.

## 5. Conclusions

This study found that rates of anastomotic leakage and overall postoperative complications did not differ significantly between the laparoscopic and open surgical approaches. Reoperations and 30-day readmissions were more frequent in the laparoscopic group, possibly reflecting more common use of the primary anastomosis. Open surgery remains relevant, particularly for more advanced cases, due to potentially superior oncologic outcomes. Laparoscopic surgery remains a viable option, especially for early-stage tumors. Usage of NOAC was linked to a higher anastomotic leakage rate while arterial hypertension unexpectedly associated with lower risk—possibly due to improved hemodynamic monitoring and optimization in these patients. Older patients with a higher ASA class (IV) exhibited a significantly higher incidence of anastomotic leakage. Given the demonstrated protective effect of diverting ileostomy against AL, increasing the use of diverting ileostomy could further reduce AL rates among high-risk patients. Future prospective studies are needed to better understand the mechanisms behind these associations and to guide evidence-based decision-making, especially in patients using anticoagulants or presenting with significant comorbidities.

## Figures and Tables

**Figure 1 medicina-61-01751-f001:**
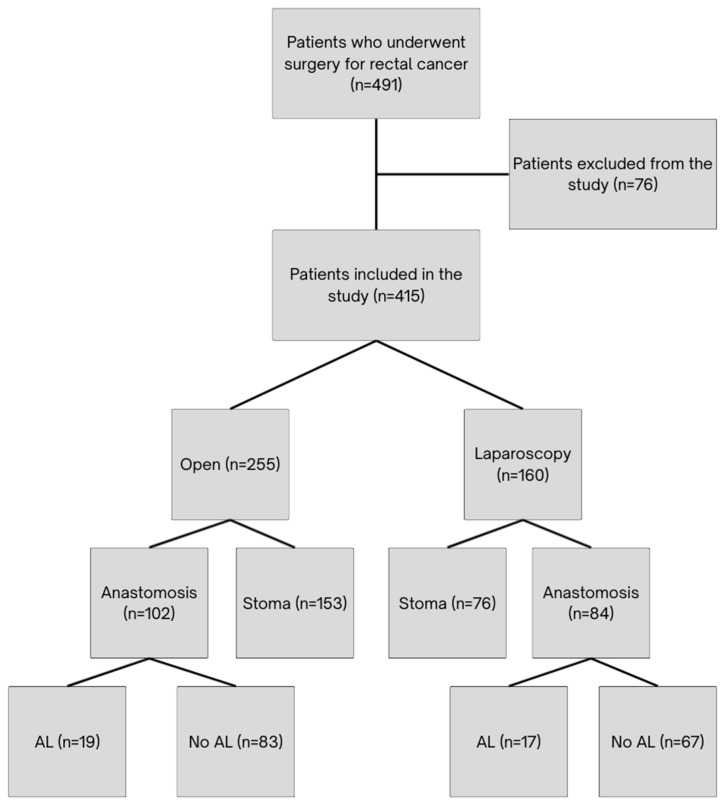
Flowchart of Study Participant Selection.

**Figure 2 medicina-61-01751-f002:**
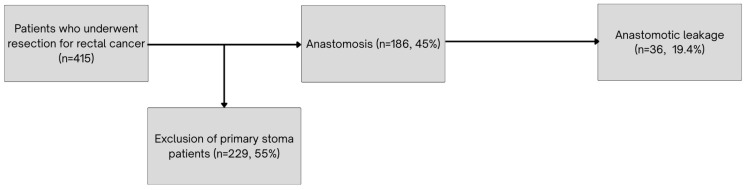
Flowchart of case selection for AL analysis.

**Table 1 medicina-61-01751-t001:** Clavien–Dindo classification of surgical complications.

Grade	Definition
I	Minor deviation, no special treatment (e.g., wound dressing).
II	Needs medication (e.g., antibiotics, transfusion).
III	Requiring surgical, endoscopic or radiological intervention.
IIIa	Intervention without general anesthesia (e.g., drainage).
IIIb	Intervention under general anesthesia (e.g., reoperation).
IV	Life-threatening complication.
IVa	Single-organ failure.
IVb	Multi-organ failure.
V	Death.

**Table 2 medicina-61-01751-t002:** Patient, Preoperative, and Perioperative Variables.

Patient Characteristics	Laparoscopy (*n* = 160)	Open (*n* = 255)	*p*
Age, median (IQR)	67 (61.0–76.7)	67 (61.0–76.0)	0.691
Sex, *n* (%)			
Male	94 (58.8)	151 (59.2)	0.925
Female	66 (41.3)	104 (40.8)	
BMI, median (IQR)	26.12 (23.45–29.64)	26.59 (23.44–30.86)	0.204
ASA score, *n* (%)			
I	0 (0)	0 (0)	0.442
II	28 (17.8)	45 (18.7)	
III	109 (69.4)	175 (72.6)	
IV	20 (12.7)	20 (8.3)	
V	0 (0)	1 (0.4)	
Neoadjuvant treatment, *n* (%)			
Yes	82 (51.2)	122 (47.8)	0.499
No	78 (48.8)	133 (52.2)	
MRF, *n* (%)			
+	69 (59.5)	120 (63.8)	0.448
−	47 (40.5)	68 (36.2)	
EMVI, *n* (%)			
+	30 (41.1)	60 (44.4)	0.642
−	43 (58.9)	75 (55.6)	
Tumor distance from the AV (cm), median (IQR)	7 (5.0–10.0)	6 (4.0–10.0)	0.124
Cancer location (anatomical subdivision), *n* (%)			
Upper rectum	58 (38.2)	73 (33.2)	0.363
Middle rectum	60 (39.5)	84 (38.2)	
Lower rectum	34 (22.4)	63 (28.6)	
Comorbidities (Arterial hypertension), *n* (%)			
Yes	110 (68.8) ^b^	147 (58.1) ^a^	**0.03**
No	50 (31.2)	106 (41.9)	
Comorbidities (Diabetes mellitus), *n* (%)			
Yes	18 (11.3)	35 (13.8)	0.458
No	141 (88.7)	218 (86.2)	
Anticoagulants, *n* (%)			
Yes	22 (13.8)	36 (14.3)	0.450
No	136 (85.5)	216 (85.7)	

(^a^, ^b^) in the same row indicate a significant difference between groups (Bonferroni test, *p* < 0.05). Bold values indicate statistically significant results.

**Table 3 medicina-61-01751-t003:** Intraoperative Outcomes.

Outcomes	Laparoscopy (*n* = 160)	Open (*n* = 255)	*p*
Type of operation, *n* (%)			
Stoma	76 (48.0)	153 (60.0)	**0.018**
Anastomosis	84 (52.0)	102 (40.0)	
Diverting ileostomy, *n* (%)			
Yes	16 (19.3)	23 (22.3)	0.611
No	67 (80.7)	80 (77.7)	
Operative time (Mean ± SD)	213.0 ± 65.9	201.3 ± 60.35	**0.050**
Pelvic drain, *n* (%)			
Yes	149 (93.1)	246 (97.2)	**0.046**
No	11 (6.9)	7 (2.8)	

Bold values indicate statistically significant results.

**Table 4 medicina-61-01751-t004:** Pathological and Postoperative Outcomes by Surgical Approach.

Outcomes	Laparoscopy (*n* = 160)	Open (*n* = 255)	*p*
(y)pT stage, *n* (%)			
0	0 (0)	3 (1.3)	**0.007**
1	18 (11.9) ^b^	7 (2.9) ^a^	
2	39 (25.8)	71 (29.8)	
3	90 (59.6)	150 (63.0)	
4	4 (2.6)	7 (2.9)	
(y)pN, *n* (%)			
0	98 (64.5)	144 (60.0)	0.804
1	42 (27.6)	72 (30.0)	
2a	8 (5.3)	15 (6.3)	
2b	4 (2.6)	9 (3.8)	
R grade, *n* (%)			
0	139 (97.2)	230 (96.6)	0.760
1	4 (2.8)	8 (3.4)	
Length of stay (days), median (IQR)	9 (7.0–13.0)	9 (8.0–12.0)	0.682
Reoperations, *n* (%)			
0	139 (86.9) ^b^	238 (93.3) ^a^	**0.027**
1	15 (9.4)	13 (5.1)	
>1	6 (3.8)	4 (1.6)	
Anastomotic leak, *n* (%)			
Yes	17 (20.5)	19 (18.4)	0.434
No	67 (79.5)	83 (81.6)	
Salvage of anastomosis, *n* (%)			
Yes	5 (29.4)	7 (36.8)	0.637
No	12 (70.6)	12 (63.2)	
Total Mesorectal Excision, *n* (%)			
Complete	47 (74.6)	102 (81.0)	0.314
Near complete	16 (25.4)	24 (19.0)	
Lymph node yield, (Mean ± SD)	11.18 ± 5.8	13.1 ± 7.8	**0.01**
Pathological lymph node yield, (Mean ± SD)	0.7 ± 1.8	1.12 ± 2.27	0.085

(^a^, ^b^) in the same row indicate a significant difference between groups (Bonferroni test, *p* < 0.05). Bold values indicate statistically significant results.

**Table 5 medicina-61-01751-t005:** Clavien–Dindo classification of postoperative complications (events during admission and 30 d thereafter).

Grade	Laparoscopy (*n* = 67)	Open (*n* = 102)	*p*
I	16 (23.9)	30 (29.4)	0.237
II	21 (31.3)	36 (35.3)	
III	1 (1.5)	2 (2)	
IIIa	0 (0)	4 (3.9)	
IIIb	24 (35.8)	24 (23.5)	
IV	2 (3)	1 (1)	
IVa	0 (0)	3 (2.9)	
IVb	0 (0)	0 (0)	
V	3 (4.5)	2 (2)	

**Table 6 medicina-61-01751-t006:** Patient and Surgical Factors Associated with Anastomotic Leakage.

Patient Characteristics	No AL (*n* = 150)	AL (*n* = 36)	*p*
Age, median (IQR)	65.0 (58.0–73.0)	64.0 (51.3–70.0)	0.138
Sex, *n* (%)			
Male,	80 (79.2)	21 (20.8)	0.292
Female	70 (82.4)	15 (17.6)	
BMI, median (IQR)	25.9 (23.1–29.9)	26.6 (24.9–30.6)	0.426
ASA score, *n* (%)			
I	0 (0)	0 (0)	**0.002**
II	41 (80.4)	10 (19.6)	
III	94 (84.0)	18 (16.0)	
IV	6 (42.9)	8 (57.1) *	
V	0 (0)	0 (0)	
Neoadjuvant treatment, *n* (%)			
Yes	59 (76.6)	18 (23.4)	0.243
No	91 (83.5)	18 (16.5)	
MRF, *n* (%)			
+	31 (81.6)	7 (18.4)	0.506
−	74 (76.3)	23 (23.7)	
EMVI, *n* (%)			
+	31 (73.8)	11 (26.2)	0.405
−	41 (80.4)	10 (19.6)	
Type of approach, *n* (%)			
Open	75 (82.4)	16 (17.6)	0.780
MIS not converted	66 (79.5)	17 (20.5)	
MIS converted	9 (75.0)	3 (25.0)	
Diverting ileostomy, *n* (%)			
Yes	35 (89.8)	4 (10.2)	0.102
No	115 (78.1)	32 (21.9)	
Operative time (Mean ± SD)	219.9 ± 62.9	229.6 ± 65.2	0.510
Pelvic drain, *n* (%)			
Yes	136 (80.0)	34 (20.0)	0.604
No	12 (85.8)	2 (14.2)	
Tumor distance from the AV (cm), median (IQR)	10 (5.7–12.2)	9 (6.5–11.5)	0.887
Cancer location (anatomical subdivision), *n* (%)			
Upper rectum	72 (81.8)	16 (18.2)	0.781
Middle rectum	50 (78.1)	14 (21.9)	
Lower rectum	16 (84.2)	3 (15.8)	
(y)pT stage, *n* (%)			
0	1 (100.0)	0 (0.0)	0.939
1	12 (75.0)	4 (25.0)	
2	43 (81.1)	10 (18.9)	
3	85 (80.2)	21 (19.8)	
4	1 (100.0)	0 (0.0)	
(y)pN, *n* (%)			
0	94 (83.2)	19 (16.8)	0.424
1	41 (76.0)	13 (24.0)	
2a	6 (67.7)	3 (33.3)	
2b	2 (100.0)	0 (0.0)	
R grade, *n* (%)			
0	139 (81.9)	31 (18.1)	0.637
1	1 (100.0)	0 (0.0)	
Total Mesorectal Excision, *n* (%)			
Yes	54 (76.1)	17 (23.9)	0.321
No	9 (90.0)	1 (10.0)	
Length of stay (days), median (IQR)	8 (7.0–9.0)	17 (12.0–23.7)	**0.000**
Reoperation, *n* (%)			
Yes		33 (91.7)	
No		3 (8.3)	
Comorbidities (Arterial hypertension), *n* (%)			
Yes	89 (85.6)	15 (14.4)	**0.05**
No	61 (74.4)	21 (25.6)	
Comorbidities (Diabetes mellitus), *n* (%)			
Yes	13 (76.5)	4 (23.5)	0.209
No	137 (81.1)	32 (18.9)	
Anticoagulants, *n* (%)			
NOAC	4 (50.0) *	4 (50.0)	**0.026**
Aspirin	7 (100.0)	0 (0.0)	
Clopidogrel	1 (100.0)	0 (0)	
No	138 (81.7)	31 (18.3)	

* *p* < 0.05 indicates a statistically significant difference. Bold values indicate statistically significant results.

**Table 7 medicina-61-01751-t007:** Multivariate Regression Analysis of Factors Associated with Anastomotic Leak.

Predictive Factor	Odds Ratio (95% Confidence Interval)	Significance (*p*-Value)
Age	1.04 (1.01–1.07)	**0.044**
Neoadjuvant treatment	1.58 (0.74–3.41)	0.238
ASA score	4.03 (1.13–4.81)	**0.022**
Diverting ileostomy	0.31 (0.10–0.99)	**0.049**
Sex (Male)	1.25 (0.57–2.71)	0.568

176 patients (140 no AL patients and 36 patients with AL) were included into logistic regression analysis. χ^2^ = 12.34, *p* < 0.05. Hosmer–Lemeshow test χ^2^ = 10.55; *p* = 0.698. Nagelkerke R^2^ = 0.106. Significance at *p* < 0.05. Bold values indicate statistically significant results.

## Data Availability

The original contributions presented in this study are included in the article. Further inquiries can be directed to the corresponding author.

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
