# Peer review of "Impact of Surgical Approach, Patient Risk Factors, and Diverting Ileostomy on Anastomotic Leakage and Outcomes After Rectal Cancer Resection: A 5-Year Single-Center Study"

_medicina, 2025, doi:10.3390/medicina61101751_

Round 1
Reviewer 1 Report
Comments and Suggestions for Authors
The authors address the influence of surgical approach, patient risk factors, and diverting ileostomy on anastomotic leakage & postoperative outcomes in rectal cancer resection. Below are my comments/questions:
- Please provide a flow diagram showing total resections to anastomoses to cases included in the AL analysis. Also, mention how many patients underwent an anastomosis, and how many had a primary stoma without anastomosis?
- Why was the outcome confined to leaks during the initial admission? Were leaks after discharge (eg. within 30 days) captured?
- How many variables did the authors included in the multivariable model? Did authors checked the events-per-variable ratio, and handling of potential overfitting? There is a mismatch in the ASA confidence interval in Table 5.
- How were patients selected for diversion?
- The rates of reoperation seem to be higher in laparoscopy despite of similar AL rates.
- Could you provide 30-day/90-day mortality, Clavien–Dindo complications, readmission, and stoma reversal rates?
Author Response
Please provide a flow diagram showing total resections to anastomoses to cases included in the AL analysis. Also, mention how many patients underwent an anastomosis, and how many had a primary stoma without anastomosis?
- As requested, we have added a flow diagram illustrating the patient selection process. The diagram outlines the total number of resections, the number of patients who underwent an anastomosis, and those who received a primary stoma without an anastomosis. Patients with a primary stoma were excluded, and the final number of cases included in the anastomotic leak analysis.
Why was the outcome confined to leaks during the initial admission? Were leaks after discharge (eg. within 30 days) captured?
- Patients with AL within 30 days postoperatively were identified through systematic monitoring of 30-day rehospitalizations. Since all patients operated in our hospital are routinely referred to our tertiary center in case of early postoperative complications, this ensures a comprehensive capture of relevant AL cases. We clarified it in Methods.
How many variables did the authors included in the multivariable model? Did authors checked the events-per-variable ratio, and handling of potential overfitting? There is a mismatch in the ASA confidence interval in Table 5.
- We thank the reviewer for this important question. In our multivariable logistic regression model, we included 5 variables that were clinically relevant and/or statistically significant in univariate analysis. The number of anastomotic leak (AL) events was 36, yielding an events-per-variable (EPV) ratio of approximately 7.2. While this is slightly below the traditional rule of thumb of 10 EPV, recent methodological studies suggest that models with an EPV of 5–9 can still provide reliable estimates, particularly when variables are carefully selected and collinearity is avoided. To minimize the risk of overfitting, we restricted the model to clinically meaningful predictors and did not include highly correlated variables. We have clarified this in the Methods section of the revised manuscript.
-Could You please clarify what do You mean mismatch in ASA CI in Table 5?
How were patients selected for diversion?
The rates of reoperation seem to be higher in laparoscopy despite of similar AL rates.
Could you provide 30-day/90-day mortality, Clavien–Dindo complications, readmission, and stoma reversal rates?
- Patients were selected for diversion based on higher risk features, including low anastomosis, neoadjuvant therapy, higher age, and advanced disease. However, we do not have any specific protocol to do standard diversion.
-The reoperation rate was higher in the laparoscopic group despite similar AL rates, most likely due to the higher proportion of patients receiving an anastomosis in this group. In addition, in some cases, a diagnostic relaparoscopy was performed when there was clinical suspicion of AL, which contributed to the slightly higher reoperation rate observed after laparoscopy.
- We appreciate the reviewer’s insightful comment. Owing to the retrospective design of our study, it was not possible to reliably capture 30-day or 90-day mortality. In particular, if a patient’s death was unrelated to the postoperative course (e.g., myocardial infarction), this information could not be systematically obtained from the national database. Similarly, stoma reversal rates could not be determined, as reversals may have been performed in other centers across the country and are therefore not traceable in our dataset.
-However, we were able to report postoperative morbidity according to the Clavien–Dindo classification and 30-day readmission rates, which are now included in the revised manuscript.
Reviewer 2 Report
Comments and Suggestions for Authors
This is a well-conducted retrospective study addressing important factors influencing anastomotic leakage after rectal cancer surgery. The sample size is adequate, and the statistical analysis is clear. The findings regarding ASA score, age, NOAC use, and the protective role of diverting ileostomy are clinically relevant. However, the relatively high leakage rate compared to multicenter trials warrants further discussion, particularly in relation to patient selection and perioperative management. The limitations section is appropriate but could benefit from additional detail on potential confounders (e.g., nutritional status, smoking). Overall, the manuscript is valuable and merits publication after minor revisions.
Comments on the Quality of English LanguageThis is a well-conducted retrospective study addressing important factors influencing anastomotic leakage after rectal cancer surgery. The sample size is adequate, and the statistical analysis is clear. The findings regarding ASA score, age, NOAC use, and the protective role of diverting ileostomy are clinically relevant. However, the relatively high leakage rate compared to multicenter trials warrants further discussion, particularly in relation to patient selection and perioperative management. The limitations section is appropriate but could benefit from additional detail on potential confounders (e.g., nutritional status, smoking). Overall, the manuscript is valuable and merits publication after minor revisions.
Author Response
We thank the reviewer for this insightful comment. As outlined in the revised Discussion, the higher anastomotic leakage rate in our cohort may be explained by the fact that our patients generally presented with more advanced tumors compared with those included in multicenter trials, which likely increased the technical complexity of surgery and the inherent risk of leakage. In addition, our institutional practice does not favor routine use of diverting ileostomies, which may have contributed to a higher proportion of clinically significant leaks. We have revised the Limitations section to acknowledge the potential influence of unmeasured confounders such as nutritional status, smoking, and other lifestyle-related factors, which were not consistently available in our retrospective dataset but may have impacted outcomes.
Round 2
Reviewer 1 Report
Comments and Suggestions for Authors
The authors have scientifically addressed all the comments.